# Elucidation of the Potential Hair Growth-Promoting Effect of *Botryococcus terribilis*, Its Novel Compound Methylated-Meijicoccene, and C32 Botryococcene on Cultured Hair Follicle Dermal Papilla Cells Using DNA Microarray Gene Expression Analysis

**DOI:** 10.3390/biomedicines10051186

**Published:** 2022-05-20

**Authors:** Aprill Kee Oliva, Meriem Bejaoui, Atsushi Hirano, Takashi Arimura, Tran Ngoc Linh, Eriko Uchiage, Sachiko Nukaga, Kenichi Tominaga, Hiroyuki Nozaki, Hiroko Isoda

**Affiliations:** 1Research and & Development Center for Tailor-Made QOL Program, University of Tsukuba, Tsukuba City 305-8550, Japan; aprillkeeoliva@gmail.com (A.K.O.); meriem.bejaoui@aist.go.jp (M.B.); 2Alliance for Research on the Mediterranean and North Africa (ARENA), University of Tsukuba, Tsukuba City 305-8572, Japan; 3AIST-University of Tsukuba Open Innovation Laboratory for Food and Medicinal Resource Engineering (FoodMed-OIL), AIST, University of Tsukuba, Tsukuba City 305-0006, Japan; takashi-arimura@aist.go.jp (T.A.); tran.linh@aist.go.jp (T.N.L.); e-uchiage@aist.go.jp (E.U.); k-tominaga@aist.go.jp (K.T.); 4Tokyo Electric Power Company Holdings, Inc., 1-1-3 Uchisaiwai-cho, Chiyoda-ku, Tokyo 100-8560, Japan; hirano.a@tepco.co.jp (A.H.); yajima.sachiko@tepco.co.jp (S.N.); nozaki.hiroyuki@tepco.co.jp (H.N.); 5Faculty of Life and Environmental Sciences, University of Tsukuba, Tsukuba City 305-8572, Japan

**Keywords:** *Botryococcus terribilis*, me-meijicoccene, C32 botryococcene, dermal papilla cells, hair growth

## Abstract

A person’s quality of life can be adversely affected by hair loss. Microalgae are widely recognized for their abundance and rich functional components. Here, we evaluated the hair growth effect of a green alga, *Botryococcus terribilis* (*B. terribilis*), in vitro using hair follicle dermal papilla cells (HFDPCs). We isolated two types of cells from *B. terribilis*—green and orange cells, obtained from two different culture conditions. Microarray and real time-PCR results revealed that both cell types stimulated the expression of several pathways and genes associated with different aspect of the hair follicle cycle. Additionally, we demonstrated *B. terribilis*’ effect on collagen and keratin synthesis and inflammation reduction. We successfully isolated a novel compound, methylated-meijicoccene (me-meijicoccene), and C32 botryococcene from *B. terribilis* to validate their promising effects. Our study revealed that treatment with the two compounds had no cytotoxic effect on HFDPCs and significantly enhanced the gene expression levels of hair growth markers at low concentrations. Our study provides the first evidence of the underlying hair growth promoting effect of *B. terribilis* and its novel compound, me-meijicoccene, and C32 botryococcene.

## 1. Introduction

The immense psychological and sociological importance of hair contributes to an individual’s outlook [1]. Hair loss greatly affects people’s quality of life despite not being an innoxious condition [1,2,3]. Low self-esteem, depression, and social and psychological distortions are only some of the consequences of hair loss [3]. Whatever the underlying cause of hair loss, it has an indisputable impact on an individual’s quality of life.

Hair is the result of coordinated processes and signals leading to cellular proliferation and differentiation within the hair follicle (HF). The HF undergoes a cycle composed of traversing various phases—anagen (growth phase), catagen (degenerative phase), telogen (resting phase), and exogen (shedding phase). However, some hair cycle disturbances can interfere with the process, causing hair loss, including premature termination of anagen, the early onset of the catagen and telogen stages, and a prolonged telogen-to-anagen transition. These abnormalities can result in the miniaturization of hair shafts in the HF and/or an increase in the number of hairs in the catagen–telogen phase [4,5,6,7,8]. These irregularities can appear in the dermal papilla.

The dermal papilla is an important element in the study of the HF niche. It is home to a population of mesenchymal cells known as hair follicle dermal papilla cells (HFDPCs) and a reservoir of multipotent stem cells to replenish the niche [9]. HFDPCs are thought to regulate HF growth and development by inducing shafts. Along with this, they stimulate several pathways such as canonical Wingless/Integrated (WNT), fibroblast growth factor (FGF), bone morrow protein (BMP), sonic hedgehog (SHH), and NOTCH [10]. The interplay of these pathways leads to HF morphogenesis.

Microalgae are photosynthetic organisms mostly found in the hydrosphere, widely spread from temperate to tropical zones. The success of microalgae and microalgae-derived substances as food, feed, fuel, and functional materials depends on thier high growth rate and undeniably rich bio-active compounds. Because microalgae are composed of a great variety of species, they come with many utilization possibilities with unknown potential [11,12,13]. The genus *Botryococcus* is a group of colonial freshwater microalgae that are known to accumulate hydrocarbons in the cells and the intercellular space [14]. Based on the type of hydrocarbons produced, there are three distinct *Botryococcus* races: Race A, B, and L. Race A produces odd numbered n-alkadienes and trienes from C_25_ to C_33_, while race B produces C_30_ to C_37_ unsaturated hydrocarbons [15]. Race L produces a single C_40_ isoprenoid hydrocarbon [16]. Since hydrocarbons are comparable with petroleum, the genus has been widely studied for renewable biofuel production [17]. However, for purposes other than fuel production there is limited information on the functionality of their extracellular and intracellular products. *B. braunii* is the most studied species in the genus for biofuel production and is also attracting attention as a source for extracting functional substances such as carotenoids [18]. In addition, an antidepressant-like effect had been found in ethanol extracts from cells of *B. braunii* [19]. Moreover, it was found that *B. braunii* shows antioxidant activity in the livers, brains, and kidneys of rats [20]. The effect of *B. braunii* in the skin was also investigated using different human skin cells and results showed that the extract reduces skin dehydration by stimulating stratum corneum hydration-related genes using keratinocytes, promotes collagen synthesis of dermal fibroblasts, and stimulates adipocyte differentiation [21]. On the other hand, *B. terribilis,* a phylogenetically closely related species to *B. braunii*, has similar physiological and morphological properties. *B. terribilis* is sometimes compared with *B. braunii* in terms of hydrocarbon productivity, but little or nothing is known about the functional substances its cells might contain. More importantly, the use of this species in hair growth research has not been analyzed to date. The elucidation of its potential effect on hair growth will be of great value as the search for natural alternatives for hair growth agents is on the rise.

This study aims to evaluate the potential hair-growth-promoting effect of *B. terribilis* using HFDPCs and to understand its molecular mechanisms. We mainly focused on the changes in the gene expression levels of the pathways associated with hair growth and the possible difference of the effect of two types of *B. terribilis* cells. We performed global gene expression profiling using DNA microarrays to elucidate the stimulated gene sets contributing to the growth of the HF. In addition, we synthesized two compounds from *B. terribilis*, namely methylated-meijicoccene (me-meijicoccene) and C32 botryococcene. These pure compounds were used to further validate the effect of *B. terribilis* on hair growth promotion and to investigate their individual contributions to this effect.

## 2. Materials and Methods

### 2.1. Sample Preparation

The TEPMO-26 strain of *Botryococcus terribilis* was isolated from the surface water of a reservoir located in Okinawa Island, Japan. The strain was isolated from the other microalgae and bacteria following the standard procedure of microalgal isolation [22]. Cultures of the strain were grown at 25 °C in culture bottles filled with a modified C medium and aerated with air containing 5% CO_2_ under continuous illumination with white LED with an intensity of 200 μmol/m^2^/s. “Green cells” were defined as cells that presented with a green color under the normal culture conditions described above; on the other hand, “orange cells” were cells that presented with an orange color after the change in the culture conditions under CO_2_ deficiency. The deficiency was achieved by introducing pure air without additional CO_2_, then green cells turned from green to orange over several days. After harvesting, both “green cells” and “orange cells” were freeze-dried and stored at −20 °C. A total of 100 mg of each sample was extracted using 70% ethanol at room temperature and placed in a dark enclosure for two weeks. Stock samples were filtered using a 0.22 μm filter unit. Green and orange cells will be referred to as BT-GC and BT-OC, respectively. Me-meijicoccene and C32 botryococcene were dissolved in DMSO and administered.

### 2.2. Cells and Cell Culture

HFDPCs were purchased from Cell Application Inc., Tokyo, Japan. The HFDPCs used were derived from the normal human hair follicles of a 56-year-old Caucasian female’s temporal scalp. HFDPCs were cryopreserved at the second or third passage. The cells can be cultured up to six passages before going into senescence with a doubling time of 30 h and a viability of 95%. The HFDPCs were maintained using Papilla Growth Medium supplemented with fetal calf serum (FBS), insulin transferrin triiodothyronine, bovine pituitary extract, and cyproterone solution (Toyobo, Japan) in a 75 cm^2^ flask (BD Falcon, London, UK). The cells were constantly monitored for media changes every two days, incubated under sterile conditions at 37 °C, and injected with 5% CO_2_ until it reached 80% confluency. Once confluent, the cells were passaged according to the culture guidelines of Cell Applications Inc. and were seeded into a new flask. HFDPCs were seeded onto a 96-well plate and 6-well plate with a density of 3 × 10^4^ for further experiments. The cells were left to attach overnight then treated for 48 h.

### 2.3. RNA Extraction

HFDPCs were seeded in a collagen-coated 6-well plate with a density of 3 × 10^4^ and left overnight to attach. Cells were treated with 1/2000 dilution of BT-GC and BT-OC, 0.1 μM minoxidil (Tokyo Chemical Industry, Tokyo, Japan) as a positive control, and 0.1 μM, 0.5 μM, 1 μM, 2.5 μM, or 10 μM of me-meijicoccene and C32 botryococcene for 48 h. After 48 h, the total RNA was extracted using ISOGEN kit (Nippon Gene, Tokyo, Japan) according to the manufacturer’s instructions and was quantified using the NanoDrop 2000 spectrophotometer (Thermo Fisher Scientific Inc., Wilmington, DE, USA).

### 2.4. DNA Microarray Analysis

Template RNA was extracted using ISOGEN solution (Nippon Gene, Japan) as described above and was used to generate biotin-labeled aRNA and then purified and fragmented. The labeled and fragmented aRNA was subjected to hybridization using the GeneChip HG-U219 Array Strip Kit (Affymetrix, Santa Clara, CA, USA). The chip was washed and stained using the Gene Atlas Fluidics Station 400 and was scanned using the Gene Atlas Imaging Station (Affymetrix, Santa Clara, CA, USA). The resulting data were normalized using the Transcriptome Analysis Console (TAC) Software (version 4.0.1). Genes with fold change ≥ 1.5 (in linear space) and *p*-value ≤ 0.05 were considered differentially expressed genes (DEGs) and were used for further analysis using the Database for Annotation, Visualization, and Integrated Discovery (DAVID). Visualization was partly conducted using R programming software.

### 2.5. TaqMan Quantitative RT-PCR Gene Expression Analysis

Based on the microarray results, the gene expression levels of hair growth-associated genes are analyzed for validation. The *B. terribilis* extracts, BT-GC and BT-OC, as well as the pure compounds, me-meijicoccene and C32 botryococcene, were subjected to gene expression analysis. The RNAs extracted as described above were used as a template for the RT-PCR analysis using Superscript IV VILO master mix (Invitrogen by Thermo Fisher Scientific, Waltham, MA, USA) per the manufacturer’s protocol. TaqMan Universal Master Mix and TaqMan probes specific to β-catenin (*CTNNB1*, Hs00355045_m1) (Applied Biosystems, Waltham, MA, USA), alkaline phosphatase (*ALPL*, Hs01029144_m1) (Applied Biosystems, Foster City, CA, USA), fibroblast growth factor-1 (*FGF1*, Hs01092738_m1), and Recombinant signal binding protein J (*RBPJ*, Hs00794653_m1) (Applied Biosystems, Foster City, CA, USA) were used. The assay was performed using 7500 Fast Real-Time PCR Software 1.3.1 (Applied Biosystems, Foster City, CA, USA) with GAPDH (Hs 02786624_g1) (Applied Biosystems, Foster City, CA, USA) as the endogenous control. All reactions were analyzed in triplicate.

### 2.6. Soxhlet Extraction

Two extracts, BT-GC and BT-OC, were prepared with 70% ethanol from carbohydrates produced by *B. terribilis* and were freeze-dried. Each sample was tested via the thin liquid chromatography (TLC) method, and the Rf values of 0.49 and 0.31 when using hexane as an eluent confirmed the presence of two main compounds, A and B. The Soxhlet extraction process is a conventional method for separating various carbohydrates from natural sources [23]. Soxhlet extraction was carried out to isolate the major components from BT-GC and BT-OC using the non-polar solvent hexane for 48 h. The hexane extract was concentrated in vacuo and the residue was placed at the top of a silica gel (Wako C-300) column and eluted with hexane. From the elute we isolated A and B as colorless oils. From 1.0 g of BT-GC or BT-OC, two components were isolated as the main compounds of each; A: 15 mg (1.5 wt%) and B: 2.5 mg (0.25 wt%). The structures of compounds A and B were determined by NMR, HRMS, and IR. The analytical instruments employed were as follows. NMR spectra were recorded on JEOL 400YH spectrometers (400 MHz for 1H NMR and 13C NMR) at room temperature (25 °C) in the Fourier transform mode. 1H NMR spectra are reported in δ units, parts per million (ppm), and were calibrated relative to the signal for residual chloroform (7.26 ppm) in chloroform-d1 (CDCl3). 13C NMR data are reported in ppm relative to CDCl3 (77.16 ppm) and were obtained with 1H decoupling. High resolution mass spectra (HR-MS) analyses were conducted using an ESI-TOF (electrospray ionization-time of flight) based on a reserpine (*m*/*z* 609.2812) matrix on a JEOL JMS-700 instrument. IR spectra were performed in attenuated total refraction (ATR) mode on a Shimadzu FT-IR QATR-S.

### 2.7. Cell Viability Assay

The cytotoxicity of me-meijicoccene and C32 botryococcene was determined by MTT assay. Briefly, HFDPCs were seeded in a collagen-coated 96-well plate with a density of 3 × 10^4^ and were left overnight to attach. The cells were treated with different concentrations (0.1 μM, 0.5 μM, 1 μM, 2.5 μM, 5 μM, 10 μM, 20 μM, 40 μM, 80 μM) for 48 h. After the treatment period, 5mg/mL MTT was added to the cells and incubated for four hours, followed by the addition of 10% sodium dodecyl sulfate (SDS) and incubation overnight. Absorbance was measured at 570 nm using a microplate reader. Cell viability was calculated as percentage (%) relative to the control or untreated cells.

### 2.8. Statistical Analysis

The results are expressed as mean ± SD. Statistical significance was determined by performing one-way ANOVA tests between the control and treated groups using GraphPad Prism 9 (San Diego, CA, USA). A *p*-value of ≤0.05 was considered significant. The results are the mean of three separate experiments.

## 3. Results

### 3.1. Comparison of Significantly Enriched DEGs of BT-GC and BT-OC

*B. terribilis* was cultured in a normal and CO_2_ deficient environment to yield two different types of cells: BT-GC and BT-OC. The study aims to differentiate whether there is a difference between BT-GC and BT-OC. Figure 1A,B shows all the DEGs within the criteria of 1.5-fold change. The red dots represent the upregulated DEGs, the green dots are the downregulated DEGs, and the gray dots represent the non-significant genes. BT-GC has a total of 6119 upregulated genes and 1469 downregulated genes. Meanwhile, BT-OC was found to have 5640 upregulated genes and 1614 downregulated genes. Figure 1C,D shows the number of DEGs in each fold change range. Most of the DEGs for both BT-GC and BT-OC treated groups showed a fold change ≥2.0 and ≤−2.0. This shows that in this range, the number of upregulated and downregulated genes for BT-GC are approximately 2700 and 700 and those of BT-OC are approximately 3800 and 700, respectively. Moreover, Figure 1E,F shows the number of genes that overlap between BT-GC and BT-OC, including 4318 upregulated and 1130 downregulated DEGs, respectively. In addition, in view of the huge number of overlapping genes and statistical analysis between BT-GC and BT-OC, it is probable that there is no significant difference between the two samples.

### 3.2. Significantly Enriched Biological Process and Cellular Components of BT-GC and BT-OC

Figure 2A,B shows the significantly enriched gene ontologies (GOs) by DEGs in the BT-GC and BT-OC treated groups. The significantly enriched GOs by DEGs of the BT-GC treated group are involved in cell adhesion, positive regulation of transcription, positive regulation of the apoptotic process, wound healing, the Wnt/β-catenin signaling pathway, the epidermal growth factor (EGF) receptor signaling pathway, positive regulation of MAP kinase activity, the Notch signaling pathway, regulation of the ERK1 and ERK2 cascades, cell growth and proliferation, and BMP pathways (Figure 2A). On the other hand, the significantly enhanced GOs by the downregulated DEGs of BT-GC include protein, phosphorylation, viral process, negative regulation of cell proliferation, RNA splicing, aging, cell cycle arrest, mitotic cell cycle, protein binding, intrinsic apoptotic signaling pathway, g2 DNA damage checkpoint, and programmed cell death (Figure 2B). BT-OC enhanced GOs like cell migration and TGF-β signaling (Figure 2C). Enhanced GOs by downregulated BT-OC are genes related to cell cycle and division, protein phosphorylation, and apoptosis (Figure 2D). Our data demonstrated that BT-GC and BT-OC showed a potential effect on hair growth by stimulating GOs that affect hair morphogenesis and the hair growth cycle.

### 3.3. Comparison of Significantly Enriched Kyoto Encyclopedia of Genes and Genomes (KEGG) Pathways by BT-GC and BT-OC

Figure 3 shows the KEGG pathways significantly enriched by DEGs. Accordingly, bubble plots were constructed to visualize the different KEGG pathways that are associated with hair growth. The bubble plot exhibits the correlation between the fold enrichment and the gene count and *p*-value for each pathway. BT-GC and BT-OC enhanced several signaling pathways related to hair growth, including the PI3K-Akt signaling pathway that has 132 and 135 genes, the Ras signaling pathway that has 82 and 75 genes, the TNF signaling pathway that has 55 and 54 genes, the p53 signaling pathway that has 34 and 30 genes, the MAPK signaling pathway that has 101 and 96 genes, and the HIF-1 signaling pathway that consists of 39 and 40 genes. BT-OC enhanced the canonical Wnt signaling pathway that consisted of 52 genes and BT-GC enhanced 42 genes related to melanogenesis (Figure 3A,C). Additionally, BT-GC and BT-OC downregulated cell cycle pathways that have 33 and 36 genes, respectively. A total of 14 genes from BT-GC and BT-OC related to RNA degradation were also downregulated. Moreover, pathways linked to lysine degradation accounted for 11 and 13 genes in BT-GC and BT-OC, respectively, and endocytosis was linked with 33 genes from both groups that were also downregulated (Figure 3C,D). Collectively, our data demonstrate that BT-GC and BT-OC stimulated a network of pathways associated with hair growth.

### 3.4. BT-GC and BT-OC Treatment Regulated Essential Functional Categories and Gene Expression Related to Hair Growth

To investigate the degree of gene expression and the possible extent of differences between BT-GC and BT-OC, we created a table summarizing the top upregulated and downregulated genes in BT-GC and BT-OC as a result of the microarray analysis. BT-GC and BT-OC enhanced the expression of CTNNB1 3.59- and 4.41-fold, respectively, compared to the control. Likewise, several genes related to the WNT pathway were enhanced. WLS was enhanced by 3.83-fold by BT-GC and 3.89-fold for BT-OC. WNT5A was stimulated by 2.36- and 3.14-fold, respectively. ALPL’s gene expression was also intensified by 2.39- and 2.37-fold by BT-GC and BT-OC, respectively. In addition, several FGF pathway-related genes are also upregulated such as FGF1, FGF12, and FGFR1 (Table 1 and Table 2). Further, RBPJ, a NOTCH pathway-related gene, was revealed to be enhanced by BT-GC and BT-OC with a fold change of 1.74-fold and 3.69-fold, respectively. On the contrary, the hair growth cycle disruption-related genes were found to be downregulated (Table 3 and Table 4). BDNF was detected to be downregulated by 3.1-fold and 1.94-fold by BT-GC and BT-OC, respectively. It was also found that DKK1, TGFB1, BMP4, and IGFBP5 were downregulated (Table 3 and Table 4).

A heatmap was generated to visualize the comparison of fold change between minoxidil, an FDA-approved drug for hair growth used here as a positive control, BT-GC, and BT-OC. Genes related to anti-inflammation, collagen, pigmentation, keratin, and hair growth were included in this heatmap. The genes related to inflammation were *PPAR-γ, PGC-1β, IL10Rβ*, and *SOCS3*. The collagen-associated genes were *COL6A1, COL6A2, COL5A2, COL1A2*, and *COL4A5*. *MITF, KRT31*, and *KRT17* were among the enhanced genes related to keratin synthesis and pigmentation. It is evident that BT-OC and BT-GC expressed a higher-fold enrichment in most of the functional categories listed as compared to minoxidil (Figure 4A). Some of the important gene markers for hair growth were chosen (*CTNNB1, ALPL, FGF1*, and *RBPJ*) and used as markers for the validation of hair growth effect. The gene expression analysis of BT-GC and BT-OC exhibited a highly significant increased expression level for *CTNNB1* (1.5-,1.6-fold), *ALPL* (1.3-,1.6-fold), *FGF1* (1.8-,2.1-fold), and *RBPJ* (1.7-,2.2-fold), respectively. The effect of the two studied samples on the gene expression of the validated makers was rather similar. (Figure 4B–E). This result confirms the positive effect of BT-GC and BT-OC through the enhancement of the gene expression levels of the essential hair growth markers on HFDPCs. Moreover, this further proves that there is no significant difference between BT-GC and BT-OC.

### 3.5. Detection and Isolation of Two Major Compounds of Botryococcus terribilis—Me-Meijicoccene and C32 Botryococcene

The most abundant compound, Compound A, was found to be a novel compound, me-meijicoccene, identified for the first time. As shown in Figure 5, the 1H NMR of me-meijicoccene was in good agreement with the previously reported 1H NMR of meijicoccene [24]. However, new peaks of the methyl group (3H, d, J = 6.8 Hz) at 1.00 ppm and a proton peak (1H, m) of the ipso position of the methyl group at 2.52 ppm appeared, leading to the identification of me-meijicoccene. Furthermore, the 1H NMR of compound B (Figure 6) was exactly the same as one of the previously reported botryococcene with 32 carbons (C32H54: C32 botryococcene) [24]. Together with other spectral data such as HRMS, they were identified as me-meijicoccene and C32 botryococcene. See Appendix A and Appendix B for the detailed characteristics of me-meijicoccene and C32 botryococcene.

### 3.6. Me-Meijicoccene and C32 Botryococcene Treatment Regulated the Gene Expression Levels of Hair Growth Markers

The cell viability assay revealed that me-meijicoccene and C32 botryococcene are not cytotoxic to HFDPCs (Figure 7A and Figure 8A). Based on this result, the optimum concentrations were chosen and used for the gene expression analysis that aimed to investigate the lowest possible effective concentration of the two compounds. Gene expression analysis was conducted to provide further validation of the effect of *B. terribilis* on hair growth. Our data demonstrated that me-meijicoccene and C32 botryococcene exhibited a significant increase in the gene expression level of the hair growth-related markers *ALPL, CTNNB1, FGF1*, and *RBPJ* after 48 h of treatment. Furthermore, the highest effect of me-meijicoccene was observed with 0.5 μm for *CTNNB1, ALPL*, and *FGF1*. RBPJ gene expression after me-meijicoccene treatment revealed a dose-dependent increase. As for C32 botryococcene, its gene expression increased in a dose dependent manner for all four gene markers (Figure 8B–D). Our data serve as the first report on the novel compound me-meijicoccene and constitute strong evidence for the hair growth effect of *B. terribilis*.

## 4. Discussion

Recently, microalgae have been gaining attention in different fields due to their sustainability, abundance, and excellent biological composition. They are known as a powerhouse of fatty acids and different metabolites such as alkaloids, terpenes, tannins, steroids, saponins, flavonoids, carbohydrates, carotenoids, lectins, halogenated compounds, mycosporine-like amino acids, and pigments [25,26]. Several of microalgae’s biological functionalities have been elucidated. This includes antioxidant, anti-inflammatory, and immunostimulating activity, as well as antifungal, antiviral, anticancer, anti-diabetes, antioxidant, antibacterial, anti-melanogenic, and anti-aging effects [27,28,29]. However, there have been no reports on *B. terribilis*’s effect on hair growth. More specifically, there has been no report on the probable effect and difference between the cells obtained from the normal and CO_2_-deficient environment. CO_2_ deficient environments can result in carotenogenesis or the production of carotenoids under severe conditions [30]. Therefore, this current study aims to provide the first evidence on the potential effect of *B. terribilis*, its novel compound me-meijicoccene, and C32 botryococcene on hair growth using human dermal papilla cells. The discovery of novel hair growth promoters could provide a superior alternative for the currently available drugs nn the market and contribute new insights to the fields of biomedicine and the pharmaceuticals.

The microarray is an excellent tool in relating the cell state to gene expression patterns in studies involving biological processes. In this study, the analysis using the global gene expression in HFDPCs in response to *B. terribilis* treatment was assessed to fully understand its potential hair growth promotion effect. BT-GC and BT-OC displayed the same trend on the number of DEGs in each range of fold changes as seen in Figure 1C,D. Furthermore, BT-GC and BT-OC share >40% of the same upregulated and downregulated DEGs (Figure 1E,F). Gene ontologies and KEGG pathways stimulated by BT-GC and BT-OC were also found to be alike. Thereby, this implies that the two cells isolated from *B. terribilis* may contain similar biological substances that trigger the enhancement of different genes.

The stimulating effect of *B. terribilis* on the canonical Wnt pathway and its associated genes (*Wls*, *Wnt5a*, and *CTNNB1)* further enhance the hair growth activity of these samples in dermal papilla cells. (Table 1 and Table 2). Wnt/β-catenin signaling plays a key role in the HF regeneration process. Stabilized β-catenin accumulation results in its binding to LEF/TCF, leading to the nuclear translocation of β-catenin-LEF/TCF. This process results in the transcription of Wnt signaling target genes and the regulation of cell proliferation and anagen initiation [31,32,33,34]. In relation to this, Wnt activators including *Lef* and *TCF1* are discovered to be positively stimulated by BT-GC and BT-OC (Table 1 and Table 2). Lef is essential for the transcriptional activation of Wnt-associated genes. On the other hand, *DKK1*, a Wnt antagonist, was found to be negatively affected by BT-GC and BT-OC. The expression of *DKK1* suppresses Wnt signaling to prevent the premature differentiation of neural precursor cells [35]. The activation of Wnt/β-catenin-promoting genes and a reduction in the expression of the negative regulators of the pathway provide important insights for the possible mechanism of action of *B. terribilis*.

Other relevant pathways were amplified including Notch signaling, ERK1/2 signaling, and MAPK signaling (Figure 2A,C). In the hair bulge, Notch regulates the cell fate of HF stem cells and controls the cell differentiation in the hair bulb [36]. *Notch* binds to *RBPJ* to impose its transactivation activities, which include hair differentiation [37]. The ERK pathway is the main pathway behind MAPK signaling [38]. These are mainly related to the cell proliferation, survival, migration, and differentiation of HFDPCs [38,39,40]. Moreover, it was found that genes associated with the FGF pathway were upregulated. *FGF1, FGF7, FGF12*, and *FGFR1* were listed. *FGF1* stimulates dermal papilla proliferation, promotes the cell cycle, and facilitates hair regeneration. *FGF7* acts on the promotion of the anagen phase and is responsible for HF morphogenesis and the regulation of the hair growth cycle. *FGF12* was found to be associated with several pathways such as the MAPK pathway, Ras pathway, and PI3k-Akt pathway. [41,42,43]. Additionally, both extracts increased the gene expression of *ALPL*, a prominent dermal papilla growth marker [44,45]. KEGG pathway analysis further determined several signaling pathways essential for hair growth and development such as TNF, Ras, PI3KT/AKT, P53, and melanogenesis (Figure 3A–D). The PI3KT/AKT signaling pathway is crucial for de novo HF regeneration [46]. TNF signaling was discovered to play a role in the formation and/or function of hair placodes and follicles [47]. Ras signaling is responsible for the proper development of the HF and the epidermis [48].

The hair cycle is an important biological clock orchestrated by processes such as the cell cycle. In parallel to this, we observed the downregulated gene ontologies associated with aging, cell cycle arrest, and cell death (Figure 2B,D). Cell cycle progression within the HF niche is an essential process for keeping the cycle going. We demonstrated the downregulation of genes related to the G1/S and G2/M transition. We speculate that *B. terribilis* prolongs the early anagen phase by slowing down the cell cycle progression through blocking the transition of cells from the G1 to the S phase [49], thus delaying the entry to the catagen phase of the hair cycle.

Androgenic alopecia and alopecia areata are characterized by a sustained follicular inflammation that leads to hair loss. A heatmap showing the relative expression intensity (Figure 4A) reveals that several pro-inflammatory genes were downregulated after the treatment of *B. terribilis*. Thus, the suppression of the pro-inflammatory genes by *B. terribilis* strengthen the effect of these samples on stimulating hair growth and preventing hair shaft shredding [50]. Moreover, keratin, collagen, and pigmentation-associated genes were enhanced along with the important hair growth markers. HFDPCs communicate closely with keratinocytes and melanocytes. The crosstalk between the three cells is regulated by secreted factors. HFDPCs affect melanocytes’ biology and differentiation, which leads to its migration to the outer root sheath to generate mature melanocytes in the hair matrix and hence hair pigment [51]. The epithelial–mesenchymal interaction between HFDPCs and keratinocytes is essential for the growth and development of the HF [52]. Additionally, the extracellular matrix (ECM), which is rich in collagen protein, plays an important role in the cell adhesion and interaction of HFDPCs [53]. Due to the promising result of the microarray analysis, we validated the results using the gene expression analyses of four important genes linked to hair growth—*CTNNB1*, *ALPL*, *FGF1,* and *RBPJ.* These four markers were chosen due to their important roles in hair growth. *CTNNB1* is essential for anagen initiation, *ALPL* is widely known as a HFDPCs proliferation marker, FGF1 is involved in morphogenesis, and *RBPJ* is linked to the NOTCH pathway that regulates hair differentiation. Choosing these three markers allows us to cover different aspect of the HF cycle. The treatment of BT-GC and BT-OC strongly enhanced the gene expression levels of the t markers after 48 **h** of treatment (Figure 4C,D). Currently, minoxidil and finasteride are the only drugs on the market used to promote hair growth. However, these drugs were reported to cause unfavorable aftereffects [54,55,56]. Our results demonstrate the similar effect of minoxidil and *B. terribilis* in vitro. Our findings suggests that *B. terribilis* has a significant effect on hair growth-associated genes, has a potential anti-inflammation effect, and can be an alternative treatment for hair loss.

Despite the compelling evidence of the hair growth promotion effect of *B. terribilis* described by our study, extracts are a very complicated mixture of various substances. To identify and purify the compound that may be the main driver of this effect, HPLC analysis was performed. We have successfully isolated C32 botryococcene and the novel compound me-meijicoccene found in *B. terribilis*. C32 botryococcene is a known compound; however, this is the first time its NMR data and characteristics have been reported. Me-meijicoccene is a novel compound and to our knowledge no study has reported this compound before. C32 botryococcene and me-meijicoccene did not affect HFDPCs’ viability (Figure 7A and Figure 8A). Additionally, the two compounds could modulate the hair growth-associated genes *CTNNB1*, *ALPL*, *FGF1* and *RBPJ* significantly at a lower concentration (Figure 7 and Figure 8). Therefore, this suggest that me-meijicoccene and C32 botryococcene may be a promising agent for promoting hair growth.

## 5. Conclusions

In conclusion, given the profound effect of hair loss on an individual’s quality of life, there is a need to explore alternative therapeutic agents for inducing hair growth. In parallel to this, there is a huge demand for nature-derived agents to fill this gap. Treatment with the *B. terribilis* extracts BT-GC and BT-OC exhibited a promising hair growth effect through the enhancement of the expression of important hair growth-related genes. *B. terribilis*’ stimulation of anagen markers, downregulation of catagen inducer genes, repression of Wnt inhibitors, and supplementation of the stimulation of other pathways contributed to the hair growth promotion effect. Overall, this study provides the first report of the effect of *B. terribilis,* as well as the novel compound me-meijicoccene and C32 botryococcene, on hair growth. Clinical studies are deemed necessary to validate the observed effect of *B. terribilis*, its novel compound me-meijicoccene, and C32 botryococcene.

## 6. Patents

The data reported in this article have been used to apply for a patent under Japanese Patent Application No. 2022-036924.

## Figures and Tables

**Figure 1 biomedicines-10-01186-f001:**
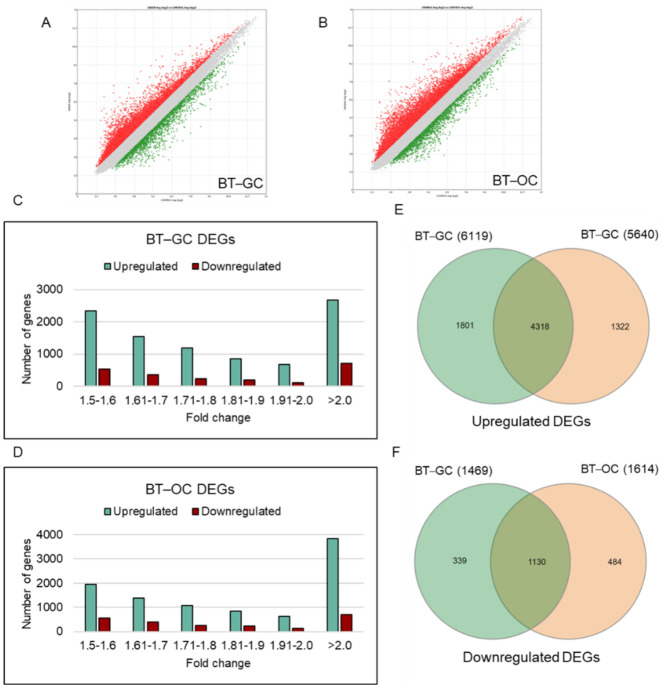
Distribution of fold changes of DEGs. (**A**) A volcano plot displaying the DEGs regulated between BT-GC and control. (**B**) A volcano plot displaying the DEGs regulated between BT-OC and control. These volcano plots were generated using Transcriptome Analysis Console 4 software. The red dots represent the upregulated genes whereas the green dots pertain to the downregulated genes. The gray dots are the unregulated genes. These DEGs are under the criteria of fold change >1.5 and <−1.5. (**C**) Distribution of DEGs regulated by BT-GC and (**D**) distribution of DEGs regulated by BT-OC. (**E**) Venn diagram showing the upregulated genes of BT-GC and BT-OC and the number of overlaps. (**F**) Venn diagram showing the downregulated genes of BT-GC and BT-OC and the number of overlapping genes.

**Figure 2 biomedicines-10-01186-f002:**
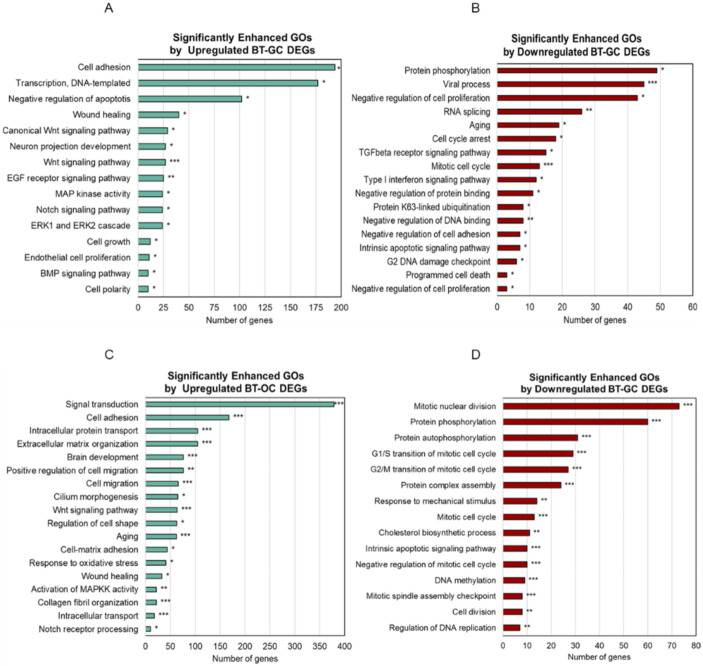
Significantly enriched biological processes of BT-GC and BT-OC DEGs. (**A**) Summary of the top significantly enriched gene ontologies (GO) by the upregulated DEGs of BT-GC. (**B**) The most significantly enriched GOs by the downregulated DEGs of BT-GC. (**C**) Summary of the most significantly enriched GOs by the upregulated DEGs of BT-OC. (**D**) The most significantly enriched GOs by the downregulated DEGs of BT-OC. Analysis was performed using the Database for Annotation, Visualization, and Integrated Discovery ver.6.8 (DAVID). * Statistically significant (*p*-value ≤ 0.05), ** statistically significant (*p*-value ≤ 0.01), *** statistically significant (*p*-value ≤ 0.001).

**Figure 3 biomedicines-10-01186-f003:**
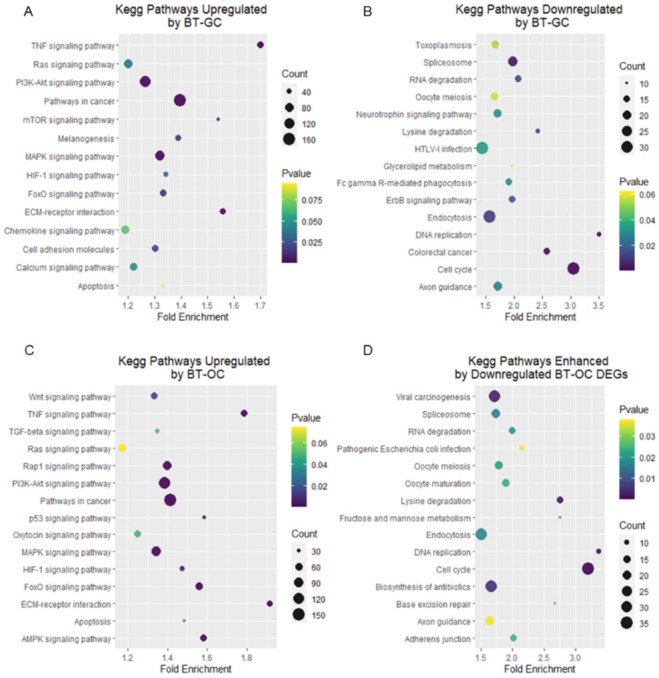
Significantly enriched KEGG pathways of BT-GC and BT-OC DEGs. (**A**) Bubble plot of the most significantly enriched KEGG pathways by the upregulated DEGs of BT-GC. (**B**) The most significantly enriched KEGG pathways by the downregulated DEGs of BT-GC. (**C**) Bubble plot of the most significantly enriched KEGG pathways by the upregulated DEGs of BT-OC. (**D**) The most significantly enriched KEGG pathways by the downregulated DEGs of BT-OC. The x-axis displays the fold enrichment, the y-axis represents the enriched pathways, the size of the bubbles pertains to the count/number of genes within the pathway, and the colors refer to the significance (*p*-value ≤ 0.05). The analysis was performed using the Database for Annotation, Visualization, and Integrated Discovery ver.6.8 (DAVID). Visualization was conducted using R programming.

**Figure 4 biomedicines-10-01186-f004:**
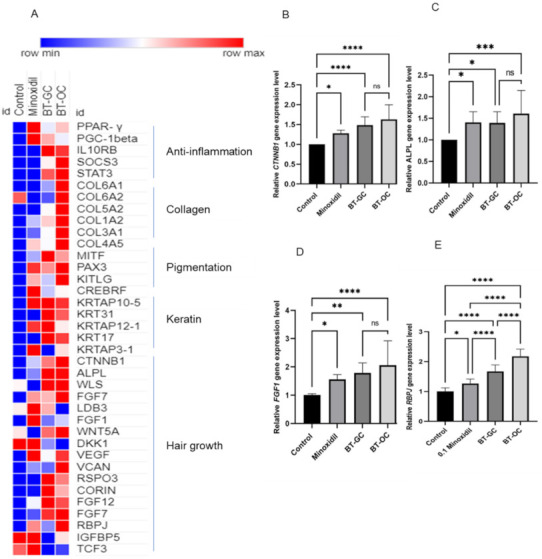
Enriched molecular functions’ categories by BT-GC and BT-OC and validation by gene expression. (**A**) Heatmap displaying the relative expression levels of DEGS. This is a comparison between BT-GC, BT-OC, minoxidil, and control. Several biological functions like anti-inflammation, collagen, pigmentation, keratin, and hair growth were chosen to be displayed using this heatmap generated by the online tool Morpheus Broad Institute. Gene expression of (**B**) *CTNNB1*, (**C**) *ALPL*, (**D**) *FGF1,* and (**E**) *RBPJ*. There is no significant difference between BT-GC and BT-OC. The result represents the mean ± SD from three independent experiments. An ANOVA test (Tukey’s multiple comparison test) was used to assess the level of significance between the groups. * Statistically significant (*p*-value ≤ 0.05), ** statistically significant (*p*-value ≤ 0.01), *** statistically significant (*p*-value ≤ 0.001), **** statistically significant (*p*-value ≤ 0.0001), ns (not significant).

**Figure 5 biomedicines-10-01186-f005:**
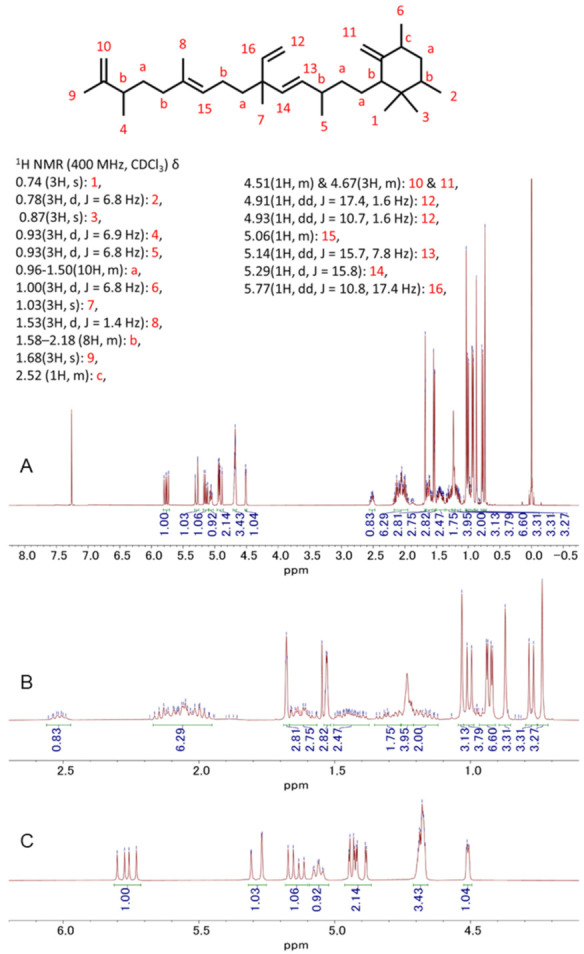
1H-NMR spectra of me-meijicoccene in CDCl3 at 25 °C: (**A**) whole spectrum; (**B**,**C**) partial spectra.

**Figure 6 biomedicines-10-01186-f006:**
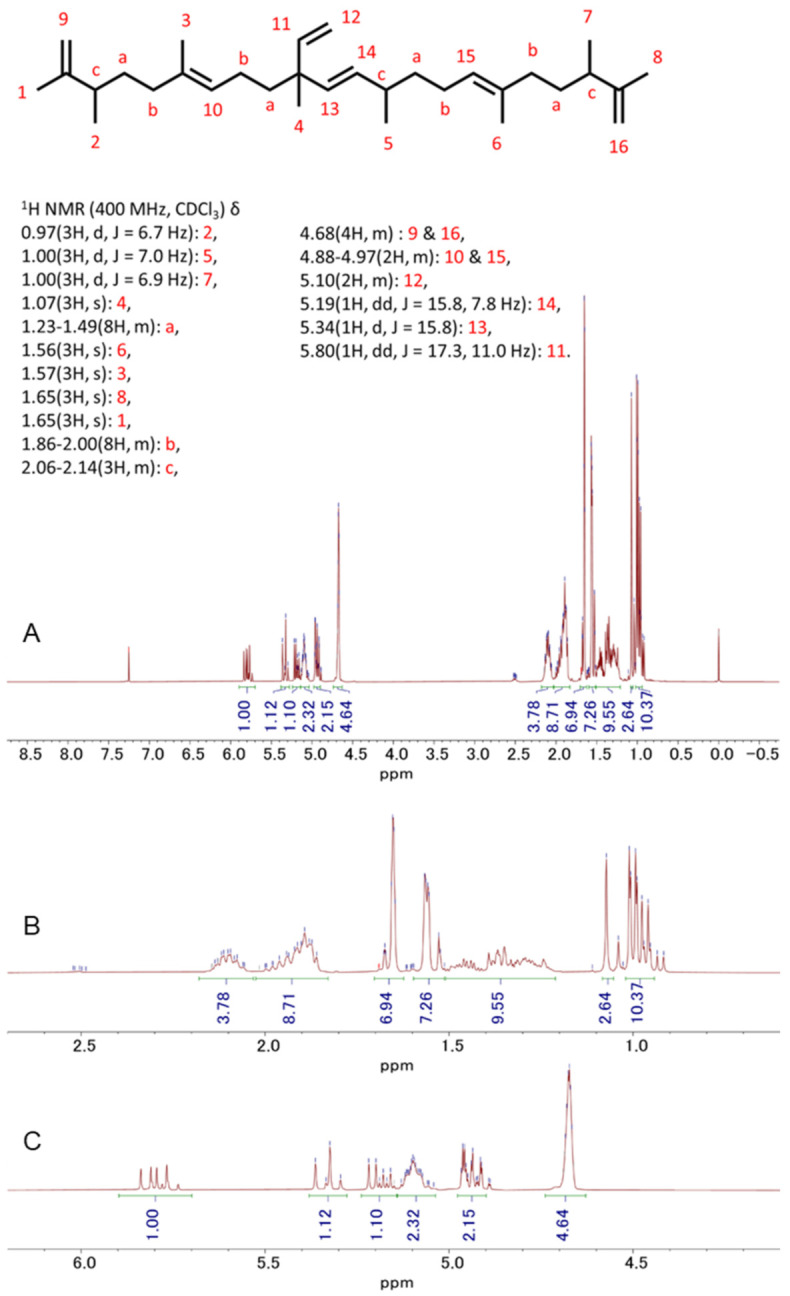
1H-NMR spectra of C32 botryococcene in CDCl3 at 25 °C: (**A**) whole spectrum; (**B**,**C**) partial spectra.

**Figure 7 biomedicines-10-01186-f007:**
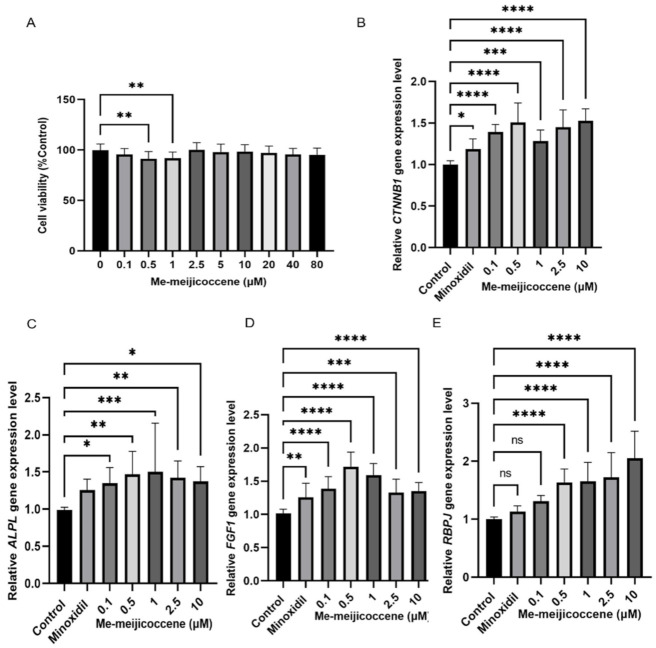
Me-meijicoccene enhanced the gene expression of hair growth-associated markers. (**A**) Determination of the relative cell viability of HFDPCs after me-meijicoccene treatment for 48 h. Validation of the relative gene expression of (**B**) *CTNNB1,* (**C**) *ALPL*, (**D**) *FGF1,* and (**E**) *RBPJ*. Results are expressed using the mean ± SD of three replicates. An ANOVA test (Dunnett’s multiple comparisons test) was used to assess the level of significance between the groups. * Statistically significant (*p*-value ≤ 0.05), ** statistically significant (*p*-value ≤ 0.01), *** statistically significant (*p*-value ≤ 0.001), **** statistically significant (*p*-value ≤ 0.0001), ns (not significant).

**Figure 8 biomedicines-10-01186-f008:**
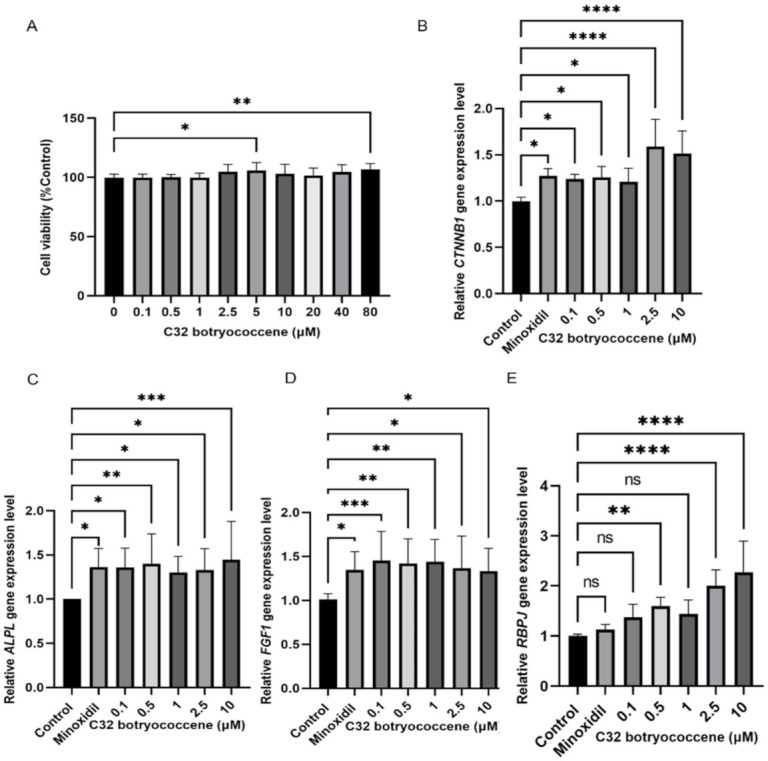
C32 botryococcene enhanced the gene expression of hair growth-associated markers. (**A**) Determination of the relative cell viability of HFDPCs after C32 botryococcene treatment for 48 h. Validation of the relative gene expression of (**B**) *CTNNB1,* (**C**) *ALPL*, (**D**) *FGF1,* and (**E**) *RBPJ*. The results are expresses using the mean ± SD of three replicates. An ANOVA test was used to assess the level of significance between the groups. * Statistically significant (*p*-value ≤ 0.05), ** statistically significant (*p*-value ≤ 0.01), *** statistically significant (*p*-value ≤ 0.001), **** statistically significant (*p*-value ≤ 0.0001), ns (not significant).

**Table 1 biomedicines-10-01186-t001:** Summary of the top upregulated hair growth-associated genes by BT-GC.

Gene Symbol	Gene Name	Biological Function	Fold-Change	* *p*-Value
*CORIN*	Corin, Serine Peptidase	Morphogenesis of HF	33.29	4.37 × 10^−20^
*SOX2*	SRY (sex determining region Y) box 2	Controls mesenchymal–epithelial crosstalk	5.31	1.72 × 10^−17^
*STAT5A*	Signal transducer and activator of transcription 5	Anagen inducer	4.2	1.25 × 10^−14^
*WLS*	Wntless	Wnt secretion and pathway	3.83	1.19 × 10^−13^
*FGF7*	Fibroblast growth factor 7	HF morphogenesis; hair growth cycle regulation	3.81	1.09 × 10^−16^
*CTNNB1*	Catenin (cadherin associated protein) beta 1	Regulation of anagen phase regulation, fibroblast growth, cell proliferation, and HF morphogenesis	3.59	1.09 × 10^−14^
*MITF*	Microphthalmia-associated transcription factor	Controls pigmentation enzymes	2.89	5.36 × 10^−12^
*ALPL*	Alkaline phosphatase	Wnt/β-catenin pathway regulator	2.39	6.93 × 10^−13^
*WNT5A*	Wingless-type MMTV, member 11	Wnt pathway	2.36	5.19 × 10^−14^
*COL3A1*	Collagen, type III, alpha 1	Fibrillogenesis in the skin and the HF	2.36	1.85 × 10^−14^
*IGFBP2*	Insulin-like growth factor Binding Protein 2	IGF1 receptor	2.17	1.31 × 10^−13^
*VCAN*	Versican	Cell aggregation, adhesion, sand proliferation	2.1	3.03 × 10^−13^
*SOX4*	SRY (sex determining region Y)-box 4	Hair regeneration	2.08	4.08 × 10^−12^
*BMP8A*	Bone morphogenetic protein 8a	Control HF growth and development	1.98	6.30 × 10^−12^
*STAT3*	Signal transducer and activator of transcription 3	Wound healing and HF development	1.9	4.92 × 10^−12^
*FGF22*	Fibroblast growth factor 22	Cutaneous development and repair	1.82	4.59 × 10^−11^
*RBPJ*	Recombinant signal binding protein J	Transcriptional effector of Notch pathway	1.74	3.51 × 10^−09^
*IGF1*	Insulin-like growth factor 1	Maintains anagen phase and cell proliferation	1.7	1.09 × 10^−08^
*LEF1*	Lymphoid enhancer factor 1	Regulation of DPCs proliferation by Wnt signaling	1.58	1.44 × 10^−09^
*HGF*	Hepatocyte growth factor	Promotes follicular growth	1.57	1.38 × 10^−08^
*FGF1*	Fibroblast growth factor 1	Anagen inducer	1.53	7.67 × 10^−07^
*FGFR1*	Fibroblast growth factor receptor 1	Primary transducers of FGF signaling in DP	1.53	1.82 × 10^−08^

* ANOVA was performed to assess the level of significance between groups. The gene expression was considered significant when the fold change was ≥1.5-fold (BT-GC versus control).

**Table 2 biomedicines-10-01186-t002:** Summary of the top upregulated hair growth-associated genes by BT-OC.

Gene Symbol	Gene Name	Biological Function	Fold-Change	* *p*-Value
*VCAN*	Versican	Regulates Cell aggregation, adhesion, and proliferation	5.46	5.66 × 10^−18^
*CORIN*	Corin, Serine Peptidase	Morphogenesis of HF	22.08	1.84 × 10^−19^
*CTNNB1*	Catenin (cadherin associated protein), beta 1	Regulation of anagen phase regulation, fibroblast growth, cell proliferation, and HF morphogenesis	4.41	2.04 × 10^−15^
*FGF7*	Fibroblast growth factor 7	HF morphogenesis; hair growth cycle regulation	4.09	6.16 × 10^−17^
*WLS*	Wntless	Wnt secretion and pathway	3.89	1.06 × 10^−15^
*COL3A1*	Collagen, type III, alpha 1	Fibrillogenesis in the skin and the HF	3.75	1.43 × 10^−16^
*RBPJ*	Recombinant signal binding protein J	Transcriptional effector of Notch pathway	3.69	1.20 × 10^−13^
*WNT5A*	Wingless-type MMTV, member 11	Wnt pathway	3.14	1.39 × 10^−16^
*SOX2*	SRY (sex determining region Y) box 2	Controls mesenchymal-epithelial crosstalk	2.95	2.74 × 10^−15^
*STAT5A*	Signal transducer and activator of transcription 5	Anagen inducer	2.81	4.48 × 10^−13^
*IGF1R*	Insulin-like growth factor 1	Maintains anagen phase and cell proliferation	2.51	1.41 × 10^−13^
*IGFBP2*	Insulin-like growth factor Binding Protein 2	IGF1 receptor	2.44	2.73 × 10^−14^
*ALPL*	Alkaline phosphatase	Wnt/β-catenin pathway regulator	2.37	7.61 × 10^−13^
*MITF*	Microphthalmia-associated transcription factor	Controls pigmentation enzymes	2.31	5.36 × 10^−11^
*STAT3*	Signal transducer and activator of transcription 3	Wound healing and HF development	2.09	1.01 × 10^−12^
*LEF1*	Lymphoid enhancer factor 1	Regulation of DPCs proliferation by Wnt signaling	2.06	8.88 × 10^−12^
*SOX4*	SRY (sex determining region Y)-box 4	Hair regeneration	1.97	1.86 × 10^−13^
*CDH3*	Cadherin 3, Type 1, P-cadherin (placenta)	Cell–cell communication and adhesion	1.95	2.73 × 10^−10^
*FGF1*	Fibroblast growth factor 1	Anagen inducer	1.94	6.17 × 10^−11^
*BMP8A*	Bone morphogenetic protein 8a	Control HF growth and development	1.76	4.94 × 10^−11^
*FGF22*	Fibroblast growth factor 22	Cutaneous development and repair	1.63	5.04 × 10^−10^
*HGF*	Hepatocyte growth factor	Promotes follicular growth	1.58	2.98 × 10^−08^
*FGFR1*	Fibroblast growth factor receptor 1	Primary transducers of FGF signaling in DP	1.52	2.16 × 10^−08^

* ANOVA was performed to assess the level of significance between groups. The gene expression was considered significant when the fold change was ≥1.5-fold (BT-OC versus control).

**Table 3 biomedicines-10-01186-t003:** Summary of the top downregulated hair growth- associated genes by BT-GC.

Gene Symbol	Gene Name	Biological Function	Fold-Change	* *p*-Value
*IGFBP5*	Insulin like growth factor binding protein 5	Negative regulator of cell proliferation in the HF	−13.19	1.43 × 10^−18^
*BDNF*	Brain-derived Neurotrophic Factor	Catagen induction and inhibition of hair shaft elongation	−3.1	2.59 × 10^−15^
*TGFB1*	Transforming growth factor beta 1	Catagen induction marker	−2.61	2.11 × 10^−13^
*ZYX*	Zyxin	Inhibits HF growth by regulating fibroblast HF cycle and promotes cell apoptosis, thus reduced expression enhances hair shaft, delays catagen entry, and HFDPCs proliferation and inductivity	−2.42	1.78 × 10^−12^
*DKK1*	Dickkopf Wnt signaling pathway inhibitor 3	Wnt signaling negative regulator	−2.35	4.81 × 10^−12^
*BMP4*	Bone morphogenetic protein 4	Suppress proliferation in the hair matrix	−2.31	3.11 × 10^−10^
*GSK3B*	Glycogen synthase kinase 3 beta	Phosphorylation of β-catenin	−1.64	1.74 × 10^−08^
*AEBP1*	AE binding protein 1	Regulator of telogen HF	−1.59	4.84 × 10^−11^
*H2AFV*	H2A histone family	Stem cell quiescence, Anagen delay	−1.53	2.06 × 10^−09^

* ANOVA was performed to assess the level of significance between groups. The gene expression was considered significant when the fold change was ≥1.5-fold (BT-GC versus control).

**Table 4 biomedicines-10-01186-t004:** Summary of the top downregulated hair growth-associated genes by BT-OC.

Gene Symbol	Gene Name	Biological Function	Fold-Change	* *p*-Value
*Igfbp5*	Insulin-like growth factor binding protein 5	Negative regulator of cell proliferation in the hair follicle	−5.29	2.63 × 10^−13^
*DKK1*	Dickkopf WNT signaling pathway inhibitor 3	Wnt signaling negative regulator	−3.67	2.33 × 10^−14^
*TGFB1*	Transforming growth factor beta 1	Catagen induction marker	−2.94	5.27 × 10^−14^
*ZYX*	Zyxin	Inhibits HF growth by regulating fibroblast HF cycle and promotes cell apoptosis	−2.37	5.64 × 10^−12^
*BDNF*	Brain-Derived Neurotrophic Factor	Catagen induction and inhibition of hair shaft elongation	−1.94	1.26 × 10^−13^
*GSK3B*	Glycogen synthase kinase 3 beta	Phosphorylation of β-catenin	−1.71	3.19 × 10^−09^
*LDB3*	LIM Domain Binding 3	Wnt-responsive gene; Regulation of hair follicle during telogen	−1.53	5.35 × 10^−06^

* ANOVA was performed to assess the level of significance between groups. The gene expression was considered significant when fold change was ≥1.5-fold (BT-OC versus control).

## Data Availability

The supporting data of this article can be found within the paper. The microarray data have been deposited in the NCBI GEO database. Expression data from Green cells (BT-GC) -treated HFDPCs. Available online: https://www.ncbi.nlm.nih.gov/geo/query/acc.cgi?acc=GSE199648 (accessed on 1 April 2022). Expression data from Orange cells (BT-OC) -treated HFDPCs. Available online: https://www.ncbi.nlm.nih.gov/geo/query/acc.cgi?acc=GSE199651(accessed on 1 April 2022).

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
