# Peer review of "Elucidation of the Potential Hair Growth-Promoting Effect of Botryococcus terribilis, Its Novel Compound Methylated-Meijicoccene, and C32 Botryococcene on Cultured Hair Follicle Dermal Papilla Cells Using DNA Microarray Gene Expression Analysis"

_biomedicines, 2022, doi:10.3390/biomedicines10051186_

Round 1
Reviewer 1 Report
The study was carried out to assess the preventative action of Botryococcus terribilis extracts against hair loss by examining its influence on gene expression in human follicle dermal papilla cells using micro-array analysis (HFDPC). Two types of extracts were tested: one green extract from normal Botryococcus terribilis cells and one orange extract from CO2 deprived cells (certainly due to the increased production of carotenoids as widely described in the literature, but not mentioned by the Authors). RT-qPCR was used to assess the expression of selected genes, and two compounds were isolated (or synthesized? - unclear) to investigate their potential role in the preventative action of the two extracts.
There are many approximations in the witting, necessitating extensive English editing. Use italics for Latin names such as plant species throughout the ms.
The description of the Materials and Methods is inappropriate, vague, and lacking in detail (first example (others hereafter), line 165 what are “hydrocarbons”?). The Authors' Instructions are not followed in this section.
Line 104: "sample is dissolved in 1 ml of 70% ethanol," do you mean extracted (not dissolved, here)?
The abstract says "we successfully synthesized," yet no synthesis description is provided. Are you referring to being isolated? It plainly alters the current paper's purpose and possible application.
How did you determine the "two major compounds" were present? The description of the isolation procedure is weird. The two extracts are prepared with 70% ethanol (polar solvent), while hexane (apolar solvent) was used to purify the "two major components" thought to be active!
When comparing gene expression data in Figures 4, 6, and 7, no increase in activity was detected with the purified (or synthesized?) compounds, which is unusual for compounds that are expected to be active. However, given the isolation approach employed, I was not shocked. It is also possible that a synergistic effect may occur, although this was not investigated.
The quality of the 1H-NMR spectra is low and did not allow any verification.
Taking all of these elements, I believe that this study is unsuitable for publishing in a journal such as Biomedicines.
Author Response
Response to Reviewer 1
General Comment: The study was carried out to assess the preventative action of Botryococcus terribilis extracts against hair loss by examining its influence on gene expression in human follicle dermal papilla cells using microarray analysis (HFDPC). Two types of extracts were tested: one green extract from normal Botryococcus terribilis cells and one orange extract from CO2 deprived cells (certainly due to the increased production of carotenoids as widely described in the literature, but not mentioned by the Authors). RT-qPCR was used to assess the expression of selected genes, and two compounds were isolated (or synthesized? - unclear) to investigate their potential role in the preventative action of the two extracts.
Response: We thank the reviewer for the comment. The reviewer is correct. The reason behind the “orange cells” is due to the CO2 deficient environment where the algae was grown. CO2 deficient environment can result to carotenogenesis or the production of carotenoids under severe conditions, hence BT-OC. Please see this explanation on p.22, Line 446-448.
Comment 1: There are many approximations in the witting, necessitating extensive English editing. Use italics for Latin names such as plant species throughout the ms.
Response 1: We thank the reviewer for this comment. We went through the entire manuscript to eliminate grammatical mistakes and confirmed that the Latin names of the plant species are written correctly.
Comment 2: The description of the Materials and Methods is inappropriate, vague, and lacking in detail (first example (others hereafter), line 165 what are “hydrocarbons”?). The Authors' Instructions are not followed in this section.
Response 2: We thank the reviewer for these comments. "Hydrocarbons" are temporarily used as all organic material produced by algae that have not undergone an extraction process with 70% ethanol. The reason for this is to isolate all non-polar carbohydrates produced by the algae and to determine their content and structure in advance. The description of Soxhlet Extraction in the reviewed manuscript shows that 427 mg of methyl-meijicoccene and 43 mg of C32 botryococcene were isolated from 1 g of algal products “hydrocarbons”. We are very sorry for the confusion caused by this description.
We described a more detailed procedure and condition for the isolation of two compounds from BT-GC and BT-OC, which were prepared with 70% ethanol, in the paragraph for “Soxhlet Extraction” in the Method section.
Comment 3: Line 104: "sample is dissolved in 1 ml of 70% ethanol," do you mean extracted (not dissolved, here)?
Response 3: We thank you the reviewer for the kind reminder. We agree, we meant “extracted” and not “dissolved” in this part. We have made the revisions accordingly.
Comment 4: The abstract says, "we successfully synthesized," yet no synthesis description is provided. Are you referring to being isolated? It plainly alters the current paper's purpose and possible application.
Response 4: Thank you for your comment. We agree with this comment. Thus, we have replaced the term “synthesized” with “isolated” in the abstract.
Comment 5: How did you determine the "two major compounds" were present? The description of the isolation procedure is weird. The two extracts are prepared with 70% ethanol (polar solvent), while hexane (apolar solvent) was used to purify the "two major components" thought to be active!
Response 5: We thank the reviewer for the question. The Rf values, 0.49 (methyl-meijicoccene) and 0.31 (botryococcene) on TLC using BT-GC and BC-OC as hexane eluent confirmed the presence of the two main compounds.
As for the description of the isolation, the answer will be the same as the first question.
The reason why two extracts, BT-GC and BT-OC, prepared with 70% ethanol (polar solvent), contain non-polar material is as follows. 70% ethanol extraction from algal products extracts not only polar carbohydrates, but also a lot of amphiphilic substances with long-chain alkyl groups. The micelles formed by these amphiphiles would take up and extract the non-polar carbohydrates.
Comment 6: When comparing gene expression data in Figures 4, 6, and 7, no increase in activity was detected with the purified (or synthesized?) compounds, which is unusual for compounds that are expected to be active. However, given the isolation approach employed, I was not shocked. It is also possible that a synergistic effect may occur, although this was not investigated.
Response 6: Thank you for your question. As mentioned, the effect of the pure compounds was not very high, this due to the lower concentration that we used as we wanted to use the minimum effective concentration. Actually, our first objective is first to discover the pure compounds responsible for the effect of B. terribilis, but after the isolation, we found two new novel compounds, so we wanted to check their effect separately, with a lower concentration, and to further validate the effect of the extract which it was our main interest at the beginning. We understand that further study to discover other pure compounds, and their synergic effect is important to further confirm our results. It can be of interest of future purposes.
Comment 7: The quality of the 1H-NMR spectra is low and did not allow any verification.
Response 7: We thank the reviewer for the comment. The partial magnified 1H-NMR spectra (b) and (c) of chemical shifts were added to the original whole 1H-NMR (a) to increase high resolution in Figure 5 & 6.
Reviewer 2 Report
Journal: Biomedicines
Article: Article: "Elucidation of the hair-loss preventing effect of Botryococcus terriblis, its novel compound —Methylated-Meijicoccene, and C32 Botryococcene using DNA microarray gene expression analysis".
Line 3 and 393: Correct misspelling in the scientific name of the alga.
Line 21: make spaces between (B.terribilis). check across the abstract.
Line 21 and 408: "in vitro"; write in italic form. check across the manuscript.
Line 63: "….algae and is well known to accumulate hydrocarbons". Which type of hydrocarbons.
Line 87: " Methylated-Meijicoccene (Me-Meijicoccene) and C32 Botryococcene"; check to write compounds names in small letter. Check across the manuscript.
Line 106: "Me-Meijicoccene and C32 Botryococcene are also dissolved in DMSO". This needs more details, how done the extraction method.
Line: 363: give the chemical class of the isolated compounds.
Author Response
Response to Reviewer 2
Comment 1: Line 3 and 393: Correct misspelling in the scientific name of alga
Response1: We thank the reviewer for this comment. We have checked and modified the misspelling of the alga’s name throughout the manuscript.
Comment 2: Line 21: make spaces between (B.terribilis). check across the abstract.
Response 2: We thank the reviewer for this observation. We have checked the abstract and have modified it according to the reviewer’s comment.
Comment 3: Line 21 and 408: “in vitro”; write in italic form. Check across the manuscript.
Response 3: We thank the reviewer for the comment. We have made the necessary modifications in Line 21 and 408 by italicizing “in vitro”.
Comment 4: Line 63: “…algae and is well known to accumulate hydrocarbons”. Which type of hydrocarbons.
Response 4: We thank the reviewer for the question. Botryococcus genus can be classified into 3 races depending on the type of hydrocarbons they produce. Race A produces C23 to C33 odd-numbered n-alkadienes, mono-, tri-, tetra-, and pentaenes, Race B produces C30 to C37 unsaturated hydrocarbons also called as “botryococcenes”, and lastly, Race L which produces a single C40 isoprenoid hydrocarbon. We have elaborated on this concern in our manuscript (p.2 Lines 64-68).
Comment 5: Line 87: “Methylated-Meijicoccene (Me-Meijicoccene) and C32 Botryococcene”; check to write compounds names in small letters. Check across the manuscript.
Response 5: We thank the reviewer for the comment. We understand this concern and have made the necessary modifications throughout the manuscript.
Comment 6: Line 106: “Me-Meijicoccene and C32 Botryococcene are also dissolved in DMSO”. This needs more details, how done the extraction method.
Response 6: We thank the reviewer for this comment. As you mentioned, “Me-meijicoccene and C32 botryococcene are also dissolved in DMSO" is not an appropriate expression. We rewrote to “Me-meijicoccene and C32 botryococcene were dissolved in DMSO and administered”.
We described a more detailed procedure and condition for the extraction of two compounds from BT-GC and BT-OC, which were prepared with 70% ethanol, in the paragraph for “Soxhlet Extraction” in the Method section.
In the reviewed manuscript before we rewrote, Line 165, as for “hydrocarbons”, "Hydrocarbons" are temporarily used as all organic matter produced by algae that have not undergone an extraction process with 70% ethanol. The reason for this is to isolate all non-polar carbohydrates produced by the algae and to determine their content and structure in advance. The description of Soxhlet Extraction shows that 427 mg of me-meijicoccene and 43 mg of C32 botryococcene were isolated from 1 g of algal product “hydrocarbons”. We are very sorry for the confusion caused by this description.
Comment 7: Line 363: give the chemical class of isolated compounds.
Response 7: We thank the reviewer for this comment. The isolated compounds are organic, but if we classify them, me-meijicoccene and C32 botryococcene are carbohydrates.
Reviewer 3 Report
This manuscript tries to imply the hair growth stimulating potency of two microalgae-derived extracts and compounds in hair specific mesenchymal cells, dermal papilla cells (HFPDCs), by microarray analysis and some hair growth related genes expression analysis.
The manuscript was well organized, including introduction section, and the data in results sections are presented clearly. The length of the manuscript is also appropriate. Table 1-4 summaries are consistent with previously reported hair growth or hair cycle related stimulating and suppressive genes such as versican, FGF7, and Stat5.
However, there are some major concerns as follows.
1. The tile should be changed, overstated “the hair-loss preventing effect” should be removed as this manuscript merely tested gene expression analysis of HFDPCs and no further biological data such as topical application of the compounds on mouse back skin to test their hair stimulating effect and clinical implication.
2. Although HFDPCs was purchased commercially, still basic biological information should be presented in the manuscript such as donor sex, ages, and original passages etc.
3. There is no description at all the condition for adding BT-GC and BT-OC in HFDPC cell culture, which including the number of each cells, the timing added, etc.. Without disclosure of these condition, I cannot judge appropriateness and validate the rest of all gene expression data. This is the case also for testing Me-Meijicoccene and C32 Botryococcene, without detailed condition such as concentration and timing used for microarray analysis and the reason for these conditions.
4. All data was used only HFDPCs but to confirm the effect of these compounds are specific to hair growth related reaction, at least another cell type as control should be included in the experiments and microarray analysis, for example, human dermal fibroblast cells.
5. In page13L335 (Figure1) and page14 L339(Figure2) are obviously miss typos, should be referred as Figure 5 and Figure 6, respectively. The authors should double check these simple miss typo errors all through the manuscript.
6. In Figure 6&7, although the authors insisted significant enhancement of CTNNB1, ALPL, FGF1 gene expression by two isolated compounds, actual increase were merely 1.2-15. Fold increase. Also, there was no explanation why these three representative genes were chosen among many other candidate genes for hair growth. What was the results for other such genes?
7. Please describe more detailed procedure and condition for isolation of two compounds from BT-GC and BT-OC cells in the paragrapf for "Soxhlet Extraction" in Method section.
Author Response
Response to Reviewer 3
General comments: This manuscript tries to imply the hair growth stimulating potency of two microalgae-derived extracts and compounds in hair-specific mesenchymal cells, dermal papilla cells (HFPDCs), by microarray analysis and some hair growth-related genes expression analysis.
The manuscript was well organized, including introduction section, and the data in results sections are presented clearly. The length of the manuscript is also appropriate. Table 1-4 summaries are consistent with previously reported hair growth or hair cycle related stimulating and suppressive genes such as versican, FGF7, and Stat5.
Response: Thank you very much.
Comment 1: The title should be changed, overstated “the hair-loss preventing effect” should be removed as this manuscript merely tested gene expression analysis of HFDPCs and no further biological data such as topical application of the compounds on mouse back skin to test their hair stimulating effect and clinical implication.
Response 1: We thank the reviewer for this suggestion. We have modified the title accordingly. We have removed “the hair-loss preventing effect” on the title and have changed the title to “Elucidation of the potential effect of Botryococcus terribilis, its novel compound – methylated-meijicoccene, and C32 botryococcene on cultured hair follicle dermal papilla cells using DNA microarray gene expression analysis”.
Comment 2: Although HFDPCs was purchased commercially, still basic biological information should be presented in the manuscript such as donor sex, ages, and original passages etc.
Response 2: We thank the reviewer for this comment. We have added the basic biological information with regards to HFDPCs on Cell Culture part of the Materials and Methods (p.3, Line 112-114).
Comment 3: There is no description at all the condition for adding BT-GC and BT-OC in HFDPC cell culture, which includes the number of each cells, the timing added, etc.. Without disclosure of these condition, I cannot judge appropriateness and validate the rest of all gene expression data. This is the case also for testing Me-Meijicoccene and C32 Botryococcene, without detailed condition such as concentration and timing used for microarray analysis and the reason for these conditions.
Response 3: We thank the reviewer for this comment. The concentration used for the gene expression and microarray analysis of BT-GC and BT-OC is 1/2000 dilution while for me-meijicoccene and C32 botryococcene are 0.1 μM, 0.5 μM, 1 μM, 2.5 μM and 10 μM. Please see p.3 Line 135-137 for the concentrations and treatment time (48 h). The density of cells can be seen in p.3 Line 121 which is 3x104. Lower concentrations of me-meijicoccene and C32 botryococcene were chosen to aim to investigate the lowest effective concentration of the compounds for future drug development.
Comment 4: All data was used only HFDPCs but to confirm the effect of these compounds are specific to hair growth related reaction, at least another cell type as control should be included in the experiments and microarray analysis, for example, human dermal fibroblast cells.
Response 4: We thank the reviewer for this comment. HFDPCs are specialized fibroblast-derived cells found at the base of the hair follicle. HFDPCs were used in this study because of their highly important role in the regulation of hair growth, and these cells are considered the fibroblasts that are mostly involved in the hair follicle. For this reason, we choose to work with these cells. Human dermal fibroblastic cells can also be considered for further study which aims on investigating the differentiation effect of the alga and the compounds. Also, if we want to see the effect of interaction between HFDPCs in the hair follicle and human dermal fibroblast in the skin.
Comment 5: In page13L335 (Figure1) and page14 L339(Figure2) are obviously miss typos, should be referred as Figure 5 and Figure 6, respectively. The authors should double check these simple miss typo errors all through the manuscript.
Response 5: We thank the reviewer for the keen observation. We have made the necessary modification with regards to the figure numbers.
Comment 6: In Figure 6&7, although the authors insisted significant enhancement of CTNNB1, ALPL, FGF1 gene expression by two isolated compounds, actual increase were merely 1.2-15. Fold increase. Also, there was no explanation why these three representative genes were chosen among many other candidate genes for hair growth. What was the results for other such genes?
Response 6: We thank the reviewer for this question. We have chosen CTNNB1, ALPL, and FGF1 because we wanted to cover three different aspects of the hair cycle. CTNNB1 is a known marker of hair growth that is mainly responsible for anagen induction. Moreover, ALPL is an HFDPCs proliferation marker, FGF1 is involved in hair morphogenesis, and RBPJ is linked to the NOTCH pathway which regulates hair differentiation.
Comment 7: Please describe more detailed procedure and condition for isolation of two compounds from BT-GC and BT-OC cells in the paragrapf for "Soxhlet Extraction" in Method section.
Response 7: We thank the reviewer for the comment. We described a more detailed procedure and condition for the isolation of two compounds from BT-GC and BT-OC, which were prepared with 70% ethanol, in the paragraph for “Soxhlet Extraction” in the Method section.
In the reviewed manuscript before we rewrote, Line 165, as for “hydrocarbons”, "Hydrocarbons" are temporarily used as all organic matter produced by algae that have not undergone an extraction process with 70% ethanol. The reason for this is to isolate all non-polar carbohydrates produced by the algae and to determine their content and structure in advance. The description of Soxhlet Extraction shows that 427 mg of me-meijicoccene and 43 mg of C32 botryococcene were isolated from 1 g of algal product “hydrocarbons”. We are very sorry for the confusion caused by this description.
Round 2
Reviewer 1 Report
This paper can be accepted depending on editor decision.
Author Response
General Comment: This paper can be accepted depending on editor decision.
Response: We thank the reviewer for providing valuable and insightful comments on our paper.

Reviewer 3 Report
I agree with all revisions by the authors except their response to my comment #4;
Comment 4: All data was used only HFDPCs but to confirm the effect of these compounds are specific to hair growth related reaction, at least another cell type as control should be included in the experiments and microarray analysis, for example, human dermal fibroblast cells.
Response 4: We thank the reviewer for this comment. HFDPCs are specialized fibroblast-derived cells found at the base of the hair follicle. HFDPCs were used in this study because of their highly important role in the regulation of hair growth, and these cells are considered the fibroblasts that are mostly involved in the hair follicle. For this reason, we choose to work with these cells. Human dermal fibroblastic cells can also be considered for further study which aims on investigating the differentiation effect of the alga and the compounds. Also, if we want to see the effect of interaction between HFDPCs in the hair follicle and human dermal fibroblast in the
The above authors’ response #4 can only be understood as a complete misunderstanding or intentional excuse to my comment#4, and it is not a convincing explanation. It is common knowledge of all hair biologists that HFDPCs are fibroblast origin(mesenchymal), and specialized one in hair follicle tissue and that HFDPCs have an essential role for hair follicle growth and differentiation. The question is whether the changes of gene profiling by the extracts are specific to HFDPCs, or just common to any cell types. I strongly recommend to perform additional experiment with the cell type other than HFDPCs at least for the factors indicated in Fig.7&8.
Author Response
Comment: I agree with all revisions by the authors except their response to my comment #4;
Comment 4: All data was used only HFDPCs but to confirm the effect of these compounds are specific to hair growth related reaction, at least another cell type as control should be included in the experiments and microarray analysis, for example, human dermal fibroblast cells.
Response 4: We thank the reviewer for this comment. HFDPCs are specialized fibroblast-derived cells found at the base of the hair follicle. HFDPCs were used in this study because of their highly important role in the regulation of hair growth, and these cells are considered the fibroblasts that are mostly involved in the hair follicle. For this reason, we choose to work with these cells. Human dermal fibroblastic cells can also be considered for further study which aims on investigating the differentiation effect of the alga and the compounds. Also, if we want to see the effect of interaction between HFDPCs in the hair follicle and human dermal fibroblast in the
The above authors’ response #4 can only be understood as a complete misunderstanding or intentional excuse to my comment#4, and it is not a convincing explanation. It is common knowledge of all hair biologists that HFDPCs are fibroblast origin(mesenchymal) and specialized one in hair follicle tissue and that HFDPCs have an essential role for hair follicle growth and differentiation. The question is whether the changes of gene profiling by the extracts are specific to HFDPCs, or just common to any cell types. I strongly recommend to perform additional experiment with the cell type other than HFDPCs at least for the factors indicated in Fig.7&8.
Response: We thank the reviewer for the comment. We agree that the effect of B. terribilis on other cell types would be interesting to know. However, our study was designed to focus specifically on HFDPCs, and we’d like to keep the focus intact. Despite that, here are some explanations of the probable effect on other cell types based on our available data.
Based on our microarray result using HFDPCs, we found that several fibroblast cell, melanocyte, and keratinocyte gene markers have been upregulated. Using this data, we would like to further explain the probable effect of B. terribilis on these cell lines. Firstly, our microarray data revealed the stimulation of the expression of several ECM markers which are also a common fibroblast marker: FBLN1, PDGFRA, DCN, COL1A1, COL1A2, COL5A1. For instance, FBLN1 (Fibullin-1), a major component of the ECM and is highly expressed in the skin. PDGFRA (PDGF-receptor alpha) is mainly involved in the PDGF signaling responsible for the activation of dermal fibroblast. These ECM and collagen-associated genes can be a strong base of the hypothesis that B. terribilis can have a positive effect on dermal fibroblasts.
Moreover, we also found that pigmentation related genes are also enhanced. We found that MITF, the main driver of pigmentation, along with other genes like PAX3, KITL, and CREBRF were upregulated. This could indicate that the B. terribilis also stimulate pigmentation, thus can also have a positive effect on human dermal melanocytes.
Lastly, it was revealed that keratinocyte- associated genes such as DMKN, PKP1, AQP3, CCL2, KRT17, KRTAP10-5, KRT31, KRTAP12-1, and KRTAP3-1 were upregulated. These are some of the canonical markers of keratinocytes that was revealed by our microarray analysis.
Having all this evidence, we do believe that B. terribilis has a high probable effect on fibroblast cells, melanocytes, and keratinocytes. We do respect that the reviewer wants to know whether the sample is specific or not to HFDPCs, however, we do believe that this is not covered by our study’s objectives being the first step to prove the effect of the sample on hair growth.
We recognize the importance of the reviewer’s comments, thus, we have included these points as a consideration for future study. As the interaction between the cells mentioned above is essential for hair growth cycle, it will be interesting to mimic their actual microenvironment. For the future study, a co-culture system or in vivo system will be considered. We acknowledge that these points are necessary to complete the studies regarding B. terribilis and its compounds. We do not intend to make this response an excuse to not do the additional experiments suggested by the reviewer, but we see this as a better foundation for a follow-up study.

Round 3
Reviewer 3 Report
I understand author's sincere and detailed explanation this time. I look forward to the progress of research in the future, and I recommend that the authors obtain further research data other than dermal Papilla cells.